# Arbitrary-Shaped Text Detection with B-Spline Curve Network

**DOI:** 10.3390/s23052418

**Published:** 2023-02-22

**Authors:** Yuwei You, Yuxin Lei, Zixu Zhang, Minglei Tong

**Affiliations:** 1College of Mathematics and Physics, Shanghai University of Electric Power, Shanghai 201306, China; 2College of Foreign Languages, Suzhou University, Suzhou 234000, China; 3College of Electronics and Information Engineering, Shanghai University of Electric Power, Shanghai 201306, China

**Keywords:** computer vision, scene text detection, Deformable DETR, B-Spline curve

## Abstract

Text regions in natural scenes have complex and variable shapes. Directly using contour coordinates to describe text regions will make the modeling inadequate and lead to low accuracy of text detection. To address the problem of irregular text regions in natural scenes, we propose an arbitrary-shaped text detection model based on Deformable DETR called BSNet. The model differs from the traditional method of directly predicting contour points by using B-Spline curve to make the text contour more accurate and reduces the number of predicted parameters simultaneously. The proposed model eliminates manually designed components and dramatically simplifies the design. The proposed model achieves F-measure of 86.8% and 87.6% on CTW1500 and Total-Text, demonstrating the model’s effectiveness.

## 1. Introduction

Human beings are in the Internet age, where mobile devices are fully pervasive, and there is an urgent need to extract valid data from images and videos. Unlike traditional Optical Character Recognition (OCR) with its simple background, single font, and uniform illumination, text detection in natural scenes often faces more significant challenges. These text regions appear on building facades, product packaging, and advertising slogans along with human activities, which bring the challenges of diverse scale shapes, complex background patterns, and variable illustration intensity. Text detection in natural scenes has a wide range of applications in image understanding, product search, real-time translation, and other scenarios. Following the increasing complexity of application scenes and the improvement of computer computing power, the focus of text detection gradually shifts from regular shapes to arbitrary-shaped and from traditional methods to machine learning-based methods.

Computer vision scientists have achieved good results for text detection in natural scenes. These methods can be roughly distinguished into segmentation-based methods, regression-based methods, and Detection Transformer (DETR)-based methods. The segmentation-based methods treat text detection as a semantic segmentation problem, which detects text by estimating the boundaries of text regions and finds text regions by pixel-level prediction, making the detection results more robust. However, this approach has difficulty distinguishing adjacent text instances. The drawback of segmentation-based methods is that they usually require redundant post-processing, significantly increasing the inference time. Regression-based methods treat the text detection problem as a regression problem, directly regressing the boundary coordinates of text regions. Most regression-based methods can predict the boundary coordinates of rectangular text regions, but they cannot handle curved text instances. Directly regressing text contours, regression-based methods can easily achieve great performance of regularly shaped text. However, these methods often deteriorate when faced with irregularly shaped text.

Inspired by Transformer [1] in natural language processing, Kuhn et al. propose the DETR [2] method for target detection. With the powerful inference capability of Transformer, DETR simplifies model design by eliminating the need to manually design components such as FPN (Feature Pyramid Network) [3], anchor generation, region suggestions, etc. Compared with Fast R-CNN [4], DETR suffers from long training time, slow model convergence, and cannot detect small targets well. To solve these problems, Zhu et al. propose Deformable DETR [5], which is ten times faster than DETR using the post-attentional feature map for training. As a classical target detection algorithm, Deformable DETR can only detect the rectangular contour of the target and cannot detect the irregular-shaped target. To address these problems, this paper proposes an arbitrary-shaped text detection model based on Deformable DETR called BSNet. The model uses B-Spline curves to model text contours and reconstructs text instances by predicting the B-Spline points of text regions. In summary, the main contributions are as follows:

(1) We propose BSNet, which is based on Deformable DETR that can detect arbitrary-shaped text. BSNet eliminates the need to manually design components such as FPN, anchor generation, and region proposal, which significantly simplifies the model’s design.

(2) By using B-Spline curves, the proposed method not only enhances the description precision of text regions but also reduces the required regression parameters, making the contours of text regions smoother and more accurate.

(3) We investigate the effects of different degrees of the B-Spline curve, various numbers of control points, and loss functions on the final detection results. The effectiveness of the proposed model is also verified on the challenging curved datasets CTW1500 and Total-Text.

The remaining sections of this paper are organized as follows: Section 2 introduces the related work on arbitrary-shaped scence text detection. Section 3 describes the network architecture, B-Spline encoder and loss design of BSNet. Training process, ablation Study and experimental results are discussed in Section 4. Section 5 presents the conclusion of this paper.

## 2. Related Works

Text detection in natural scenes is developed from traditional target detection, such as SSD (MultiBox Detector) [6] and YOLO [7], which detect text regions by directly regressing the bounding box coordinates of text regions. With Transformer’s great success in NLP (Nature Language Processing), More and more scholars in the field of computer vision are turning to Transformer-like approach. Vision Transformer [8] treats a picture as many small patches, then feed to a pure Transformer architecture with positional embeddings to distinguish different objects. Recent DETR-based method [9] try to introduce pure transformer architecture into scene text detection, but they have never been able to achieve satisfactory results. DPTNet [10] discard pure Transformer architecture and propose a parallel design that integrates the convolutional network with a self-attention mechanism to enhance the attention path and convolution path. By combining a traditional CNN-base backbone and Transformer codec, Raisi et al. [11] propose an end-to-end trainable architecture that achieved SOTA (State-Of-The-Art) performance. However, text regions in natural scenes are complex and variable in shape, and describing text regions by rectangles or quadrilaterals makes modeling inadequate and leads to low accuracy of text detection. Some methods attempt to model text regions using different representations, such as Bézier curves [12,13], Fourier descriptor [14], and wavelet descriptor [15]. These methods predict implicit parameters to reconstruct text boundaries, making the contours of text regions more refined and smoother. Describing the contours of the target region is an essential step in image processing and automated processes. Image processing often requires accurate descriptions of the shapes of complex targets. We can roughly divide description subsets for the shape into three categories: methods based on the spatial location relationship of contour points, transformation domain methods, and contour region-based methods. To improve the detection precision and efficiency of text regions of different sizes, almost all methods use a multi-scale feature approach when extracting visual features, i.e., using backbone networks and performing feature fusion via FPN. Traditional text detection algorithms are based on the spatial location relationship of contour points and directly use the coordinate of contour to describe the text region in the Cartesian coordinate system. TextRay [16] believes modeling with contour points in the Cartesian coordinate system ignores the global geometric constraints. They model the text contour in polar coordinates to solve this problem, encoding complex geometric layouts into unified representations. TextRay reconstructs the text region by predicting the coordinates of the text center and the length of each angle from the center point to the boundary. Unlike most approaches that model in the spatial domain, FCENet [14] uses Fourier transform to model text contours in the frequency domain. Theoretically, Fourier coefficients can fit closed curves arbitrary-shaped, which are well-suitable for describing text regions. Moreover, the shape of the text region is not overly complex, and its Fourier coefficients mainly exist on the low-frequency area, thus eliminating the need for high-frequency coefficients to participate in the transformation. It significantly reduces the number of predictable parameters. WDNet [15] decomposes text contours into wavelet coefficients, extracts feature sequences by discrete wavelet transform, and reconstructs text regions by predicting the wavelet feature sequences of text regions. TextSnake [16] describes a text region as a sequence of circles with different coordinates and radii to achieve an arbitrary-shaped text description, with the center of each circling on the text region’s central axis. By treating a text region as a collection of a bunch of circular contours, TextSnake also achieves good results. LeafText [17] believes that the existing methods based on contour point sequences achieve comparable performance, but these methods cannot cover some highly curved text instances. They combine the geometric features of text and bionics approach to design a text representation based on natural leaf veins. The text contour is considered as a leaf edge and it is represented by main veins, side veins and fine veins.

## 3. Method

The arbitrary-shaped text detection model proposed in this paper is shown in Figure 1. The left part shows a conventional CNN backbone, ConvNeXt [18], which is pre-trained on ImageNet, for initial visual feature extraction of the input image to obtain feature maps at different scales. Afterward, feed the feature maps into the Transformer encoder to extract the features of the text regions. The Transformer decoder regresses the B-Spline control point coordinates of the text contours from the feature maps of the text regions. Lastly, the B-Spline decoder reconstructs the boundary coordinates of the text regions using the B-Spline control points and obtains the final detection result after removing the repeated predictions by the non-maximum suppression algorithm (NMS).

### 3.1. Datasets Resample

The number of annotation points for a single text area is determined for CTW1500, which is annotated by 14 contour points, while the number of annotation points for Total-Text is uncertain. The number of annotation points of Total-Text varies according to the complexity of the text area. To accommodate datasets with different numbers of annotation points, the original annotation points need resampling before modeling the text contours using the B-Spline curves. The resampling strategy adopted in this paper is as follows: we divide the text outline into two parts (top and bottom), resample each part into 43 equally spaced points by interpolation algorithm, and then select one point for every 7 points, i.e., label each text area with 14 points to accommodate datasets with different numbers of ground truth. The annotations before and after resampling are shown in Figure 2.

### 3.2. B-Spline Curve Modeling

B-Spline curve is a function that can approximate a segmented polynomial of an arbitrary function, which has good support, flexibility, smoothness, and accuracy. Its definition [19] is shown in Equation (Equation 1):(1)$$u\left(t\right)=\sum_{i=0}^{n}b_{i}\left(t\right)B_{i,n}\left(t\right),0\leq t\leq 1,$$
(2)$$B_{i,0}=\left\{\begin{array}{c} 1,x_{i}\leq x<x_{i+1}\\ 0,otherwise \end{array}\right.,$$
(3)$$B_{i,n}\left(t\right)=\frac{t-t_{i}}{t_{i+n}-t_{i}}B_{i,n-1}\left(t\right)+\frac{t_{i+n+1-t}}{t_{i+n+1}-t_{i+1}}B_{i+1,n-1}\left(t\right),i=0,\cdots ,n,$$
where $b_{i}\left(t\right)$ denotes the *i*-th control point, and the $B_{i,n}\left(t\right)$ are the cubic B-Spline basis functions. The B-Spline curve degenerates to a straight line when only two endpoints exist. It is a uniform B-Spline curve when $\left[x_{i},x_{i+1}\right]$ is equally spaced, and a non-uniform B-Spline curve when the opposite is true. Based on the B-Spline curve, we transform the arbitrary-shaped text detection problem into the problem of predicting the coordinates of B-Spline control points. Given the curve boundary point $p_{i}\left(t\right)$, use the linear least squares method to find the optimal parameter in Equation (Equation 1), as shown Equation (Equation 4):(4)$$\left[\begin{array}{ccc} B_{0,3}\left(t_{0}\right) & \cdots & B_{3,3}\left(t_{0}\right)\\ B_{0,3}\left(t_{1}\right) & \cdots & B_{3,3}\left(t_{1}\right)\\ \vdots & \ddots & \vdots \\ B_{0,3}\left(t_{m}\right) & \cdots & B_{3,3}\left(t_{m}\right) \end{array}\right]\left[\begin{array}{cc} b_{x0} & b_{y0}\\ b_{x1} & b_{y1}\\ b_{x2} & b_{y2}\\ b_{x3} & b_{y3} \end{array}\right]=\left[\begin{array}{cc} p_{x0} & p_{y0}\\ p_{x1} & p_{y1}\\ p_{x2} & p_{y2}\\ p_{x3} & p_{y3} \end{array}\right],$$
where *m* denotes the number of curve boundary points, from Equations (Equation 1) and (Equation 4), we obtain the parameterized B-Spline curve.

### 3.3. Visual Feature Extraction

ConvNeXt extracts visual features from the input image to reduce the scale and obtain feature maps at different scales. Then send the feature maps at different scales to the Transformer encoder separately and set the number of layers of the encoder to 6. 1 × 1 convolutional kernel, reduces the number of channels at each scale to 256 and then expands them to a one-dimensional sequence. The Transformer encoder computes deformable attention for each scale feature map to generate new feature vectors. Unlike the traditional attention module, which needs to focus on global points, the deformable attention module only focuses on a small number of key sampling points around the reference point, regardless of the feature map size. This dramatically improves the training speed of the model. Deformable attention is shown in Figure 3, and the deformable attention calculation process is as follows:
(5)$$\mathrm{DeformAttn}\left(z_{q},p_{q},x\right)=\sum_{m=1}^{M}W_{m}\left[\sum_{k=1}^{K}A_{mqk}\cdot W_{m}^{\prime }x\left(p_{q}+\Delta p_{mqk}\right)\right],$$
where the input feature maps $x\in R^{C\times H\times W}$, $z_{q}$ and $p_{q}$ are the original features and reference points of the feature maps that have q indexes. m denotes that the attention module is an M-head attention module, K represents the number of all sampled key points. $A_{mqk}$ denotes queries that dot product of queries and keys, $W_{m}^{{}^{\prime }}x$ is the weight of the feature, $▵p_{mqk}$ denotes the offset. Where $▵p_{mqk}$ and $A_{mqk}$ both computed by $z_{q}$ through the fully connected layer. Using multi-scale feature maps for text detection usually significantly improves the detection accuracy, and the model proposed in this paper uses a multi-scale deformable attention module based on deformable attention as follows: (6)$$\mathrm{MSDeformAttn}\left(z_{q},{\hat{p}}_{q},{\left\{x^{l}\right\}}_{l=1}^{L}\right)=\sum_{m=1}^{M}W_{m}\left[\sum_{l=1}^{L}\sum_{k=1}^{K}A_{mlqk}\cdot W_{m}^{\prime }x^{l}\left(\phi_{l}\left({\hat{p}}_{q}\right)+\Delta p_{mlqk}\right)\right],$$
where L represents the number of the feature map’s layers and ${\hat{p}}_{q}$ is the normalized reference point. Since we need to calculate feature maps at multiple scales simultaneously, it is necessary to normalize the reference points of feature maps at different scales. The result of the operation is the remapped feature vector.

### 3.4. Text Region Reconstruction

Same as the encoder, set the number of layers of the decoder to 6. Each layer of the decoder has eight attention heads. The Transformer decoder initializes 300 vectors to calculate self-attention, then calculates the deformable attention with the remapped feature vectors from the Transformer encoder. After completing the above steps, we can get the B-Spline control point coordinates and the confidence score via two feed-forward neural networks. Next, we can reconstruct the text contours from the B-Spline control points. The solved text contour coordinate points with the confidence score work together to remove duplicate contours using non-maximum suppression algorithm (NMS) to obtain the final predicted region coordinates. Python open-source package NURBS-Python [20] enable B-Spline related operations.

### 3.5. Loss Function Design

Since Deformable DETR outputs the prediction results as an ensemble, it is necessary to match them with the ground truth before calculating the loss function. We will consider the unmatched prediction parts as background, and they will not participate in calculating the loss function. In this paper, we use the Hungarian algorithm [21] which deleted the box part to solve the above matching problem: (7)$$\hat{\sigma }=\underset{\sigma \in S_{N}}{argmin}\sum_{i}^{N}L_{match}\left(y_{i},{\hat{y}}_{\sigma (i)}\right),$$
(8)$$L_{match}\left(y_{i},{\hat{y}}_{\sigma (i)}\right)=-1_{\left\{c_{i}\neq ⌀\right\}}{\hat{p}}_{\sigma (i)}\left(c_{i}\right),$$

The Hungarian algorithm can find the best 1-to-1 match between the predicted regions and the ground truth by constructing an N×M cost matrix of the predicted *N* regions and the *M* ground truth regions. It computes the generalized distance between *M* labels and *N* predictions. The closer the distance represented, the closer the relationship between the two.

The loss functions of the model proposed in this paper include the classification loss function $L_{cls}$, the regression loss function $L_{reg}$ and the segmented rectangular GIoU loss function $L_{GIoU}$. So, we can optimize the whole model by the following:(9)$$L=L_{cls}+L_{reg}+L_{GIoU},$$
where the classification loss function $L_{cls}$ is a typical cross-entropy loss function: (10)$$L_{cls}=-\sum_{i=1}^{n}c_{i}log{\hat{c}}_{i},$$
where ${\hat{c}}_{i}$ is the expected output of the classification, $c_{i}$ is the actual output of the classification.

The regression loss function $L_{reg}$ directly minimizes the distance between the prediction and ground truth, N is the number of B-Spline control points, $\omega_{i}$ is the weight of each set of coordinates, and L1 denotes the smoothed-L1 loss function. $\left(x_{i},y_{i}\right)$ is the predicted B-Spline control point, and $\left({\hat{x}}_{i},{\hat{y}}_{i}\right)$ is the ground truth of the B-Spline control point:(11)$$L_{reg}\left(\left(x,y\right),\left(\hat{x},\hat{y}\right)\right)=\frac{1}{N}\sum_{i=1}^{N}\omega_{i}\cdot L1\left(\left(x_{i},y_{i}\right),\left({\hat{x}}_{i},{\hat{y}}_{i}\right)\right),$$

$L_{GIoU}$ further constrains the network parameter regression by calculating the segmented rectangular GIoU loss function for the four adjacent B-Spline control points. As shown in Figure 4, for a single-sided 6-point B-Spline, we can calculate the segmented rectangular GIoU loss function of the B-Spline control points by dividing the 12 B-Spline control points into five groups. The first rectangle uses the bounding rectangle of the quadrilateral formed by the four points C1, C2, C11 and C12; the second rectangle uses the bounding rectangle of the quadrilateral formed by the four points C2, C3, C10 and C11, and so on. Where ω denotes the weight of $L_{GIoU}$, *N* denotes the number of segmented rectangles, GIoU denotes the GIoU loss function. $b_{i}$ and ${\hat{b}}_{i}$ represent the coordinates of the predicted rectangle box and the actual rectangle box, respectively: (12)$$L_{GIoU}=\omega \sum_{i=0}^{N}GIoU\left(b_{i},{\hat{b}}_{i}\right),$$

## 4. Experiments

In this section, we first describe the training environment, parameter settings, and specific training steps of BSNet. Then introduce CT1500 and Total-Text which use for training and validation in this paper. Finally, we demonstrate our exploration of the BSNet structure through a series of ablation experiments, which demonstrate the effectiveness of the proposed model.

### 4.1. Implementation Details

By using the B-Spline curve, we propose BSNet. We select ConvNeXt [18] as the backbone and pre-train it on ImageNet [22]. ConvNeXt borrows a Transformer-like [1] training approach on ResNet50 [23] to control variables on Macro Design, Deep Separable Convolution [24], Inverse Bottleneck Layer [25], Large Kernel, and Micro Design to optimize model performance while keeping FLOPs essentially unchanged. During the training process, we rescale all training images to 800×800 and adopt data enhancement strategies such as color dithering, random scaling, random flipping, random cropping, and random rotation to improve the diversity of the input images. We build the model using PyTorch 1.13.0 framework and train it on Ubuntu 18.04 by NVIDIA Tesla V100 SXM2 32GB graphics card with batch size set to 12. We update the model parameters by AdamW as an optimizer, set the initial learning rate to 2 × 10${}^{-4}$, and adjust the learning rate using the poly strategy. Adopting the strategies above, training 1000 epochs only with the $L_{reg}$ loss function. After obtaining the initial model weights, we add the segmented rectangular GIoU loss function $L_{GIoU}$ to train another 600 epochs to complete the training.

In the testing phase, we rescale the test image size to 800×800 for CTW1500 and Total-Text. We statistically measure the recall, accuracy, and F-measure of the completed trained model on the above two publicly available datasets for text detection.

### 4.2. Datasets

To better validate the performance of the proposed model, we use two challenging text datasets in natural scenarios with arbitrary-shaped as follows:

SUCT-CTW1500 [26] is a Chinese and English curve dataset. It is widely used for arbitrary-shaped text detection. 1000 training images and 500 test images are available for CTW1500. CTW1500 marks each text region using 14 points with word level.

Total-Text [26] is a challenging dataset for arbitrary-shaped text detection containing curved, horizontal, and multi-directional text. 1255 training images and 300 test images are available for Total-Text. We label each text instance as a polygon with a word-level annotation.

### 4.3. Ablation Study

To investigate the effects of the different settings of B-Splines, and various loss functions on the detection results, we conduct ablation experiments on CTW1500. In order to study the impact of different degrees of B-Splines and control points on the detection results, we only use the $L_{reg}$ loss function to train the model. In the first two parts, the backbone of the model is ResNet50 [23] with DCN [27].

As shown in Table 1, for a certain degree of B-Splines, a single boost in the number of control points does not improve the model’s detection performance but instead increases regression parameters. The degree of B-Splines is 4, and the F-measure achieves 84.2% when the number of control points is 4. When the number of control points is 6, the F-measure of the model reaches a maximum value of 84.7%.

We use fourth-degree 6-point B-Splines to explore the impact of different loss functions on the detection results. The results are analyzed after further training 600 epochs on the pre-trained model obtained in Table 1 using different loss functions. The optimizer chosen for this part is Adam, which sets the initial learning rate to 1 × 10${}^{-4}$ and reduces it to 1 × 10${}^{-5}$ and 1 × 10${}^{-6}$ at 100 and 500 epochs, respectively.

As shown in Table 2, the F-measure of the proposed model is 84.7% when only using $L_{reg}$ to train the model. Next, we train the model using $L_{GIoU}$, the F-measure is 84.8%, a little higher than using $L_{reg}$. Finally, we use both $L_{reg}$ and $L_{GIoU}$ to train the model, the performance of the model further improves, and the F-measure reaches 85.4%, proving the loss function’s effectiveness. The segmented rectangular GIoU loss functions $L_{GIoU}$ can help the regression loss function $L_{reg}$ further constrain the approximate shape of the text regions to improve the model’s performance.

To evaluate the effectiveness of the architecture, we further replace the backbone of the proposed method. We select the most popular backbone ResNet50 with DCNv2 and ConvNeXt. Network computing complexity analysis is a very necessary part in computer vision. The community commonly uses parameters (Params) and Floating-point Operations per second (FLOPs) to measure the complexity of convolutional networks. As shown in Table 3 The Params and FLOPs are calculated for the input image of 800×800×3. After we change the backbone from ResNet50 to ConvNeXt, while Parameter and FLOPs rise just a little bit, the F-measure has a huge boost on CTW1500 and Total-Text, increaseing 1.4% and 2.6%, respectively. The F-measure of BSNet equipped with ConvNeXt on Total-Text is a greater improvement than CTW1500, we believe that the Transformer-base method is more sensitive to the number of training data. More data for training means more performance improvement, and Total-Text has 255 more images for training than CTW1500. The experiments indicate the effectiveness of ConvNeXt.

### 4.4. Evaluation

The B-Spline Curve Network converts the text detection problem into a B-Spline control points regression problem, which is extremely helpful for arbitrary shape text detection. For simple rectangular shaped text areas, the B-Spline curve usually degenerates to a straight line which only needs two end points. More control points for regular text introduce more parameters to the system and bring more uncertainty. Consequently, the method proposed in this paper is more suitable for text detection of arbitrary shapes.

We adopt the above training strategies to train the proposed model on CTW1500 and Total-Text without any additional datasets and collect the experimental results. Then adopt the recall (R), precision (P), and F-measure (F) as evaluation metrics and compare them with other arbitrary-shaped scene text detection algorithms, as shown in Table 4, where Ext denotes additional datasets. The models proposed in this paper outperform the mainstream algorithms on the multilingual curved text dataset CTW1500, with an F-measure of 86.8%. It is 2.3%, 1.5% and 1.5% higher than the mainstream arbitrary-shaped text detection algorithms DRRG [28], DBNet++ [29] and WDNet [15]. It is 0.4% higher than the best algorithm, TPSNet [30]. Best performance is also achieved on Total-Text compared to mainstream algorithms.The F-measure of the BSNet reaches 87.6% on Total-Text, which is a tie with the best performing algorithm TPSNet and much higher than other algorithms. However, it is worth noting that BSNet does not require an additional dataset during training compared to TPSNet, proving the superiority of our proposed model.

Figure 5 shows the comparison of the proposed model with ABCNet [12,13], FCENet [14], and WDNet [15] on CTW1500. ABCNet detects text by predicting two independent Bézier curves, which is more effective than other methods in regular multi-directional text. However, for dense curved text, curved text, and small text regions, ABCNet is more likely to confuse the adjacent. FCENet predicts text instances by Fourier descriptor embedding. FCENet performs well in most cases but does not perform well for large aspect ratio text. WDNet uses wavelet descriptors to describe text instances. Like FCENet, WDNet is not as precise for regular text contours, especially for the four corners of rectangular text areas, which are described by arcs and result in missing corners. The comparison shows that the proposed model can effectively detect multi-directional and curved text without any additional training data. The proposed method is robust to perspective and background interference and can also handle texts with large aspect ratios.

## 5. Conclusions

The shape of text regions in natural scenes is complex and variable. Directly predicting text contour point coordinates using neural networks cannot describe the text region accurately and smoothly. Therefore, this paper proposes a Deformable DETR-based arbitrary-shape text detection model, which predicts two sets of B-Spline control point coordinates then reconstructs the text contour coordinates. Using B-Splines to model text regions allows for smoother and more accurate text contours while reducing the number of predicted parameters. By designing the network structure and loss function, the model can better adapt to the text with complex and variable shapes. The experimental results show that the F-measure of the proposed model achieves 86.8% and 87.6% on the public data sets CTW1500 and Total-Text, respectively, without any pre-training datasets, which proves the effectiveness of the proposed model. However, the model proposed in this paper suffers from an excessive number of parameters and complex computation. In the next step, we will investigate lightweight to increase its practicality.

## Figures and Tables

**Figure 1 sensors-23-02418-f001:**
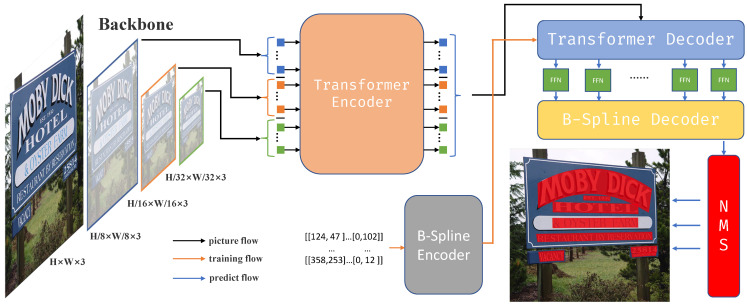
Architecture of the proposed model.

**Figure 2 sensors-23-02418-f002:**
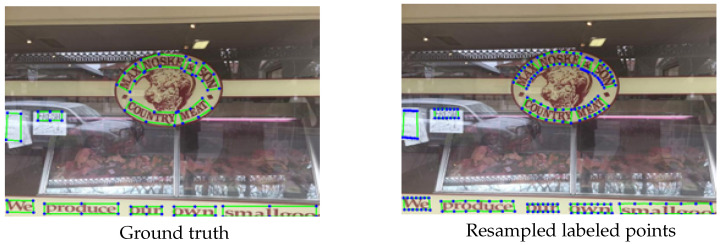
The ground truth and resampled annotation points.

**Figure 3 sensors-23-02418-f003:**
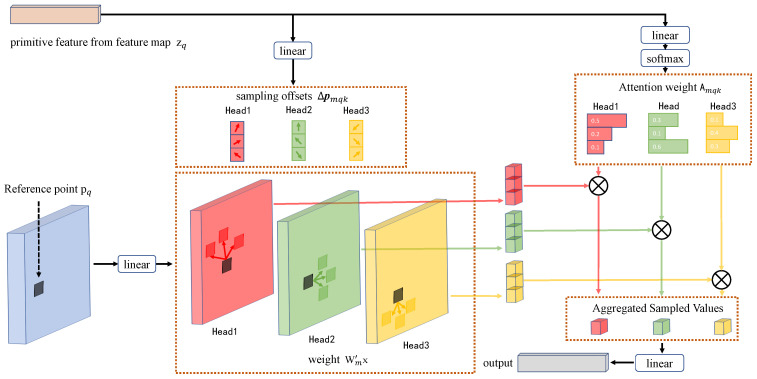
Deformable attention module.

**Figure 4 sensors-23-02418-f004:**
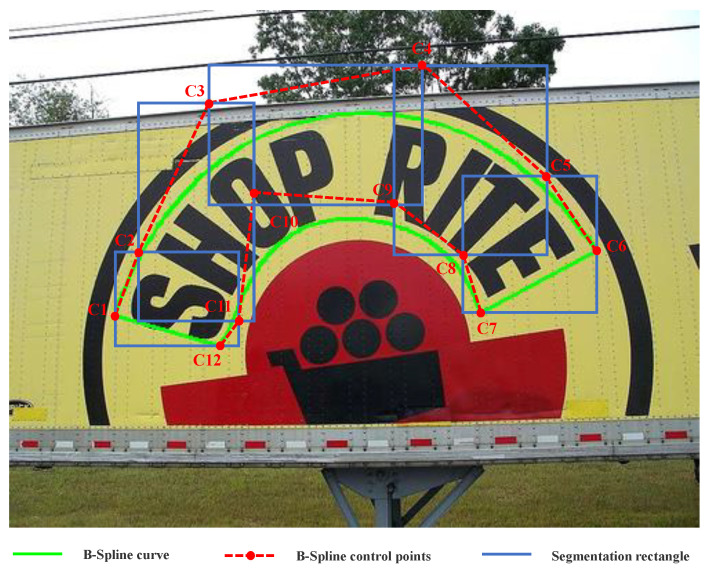
Example of segmentation rectangle GIoU.

**Figure 5 sensors-23-02418-f005:**
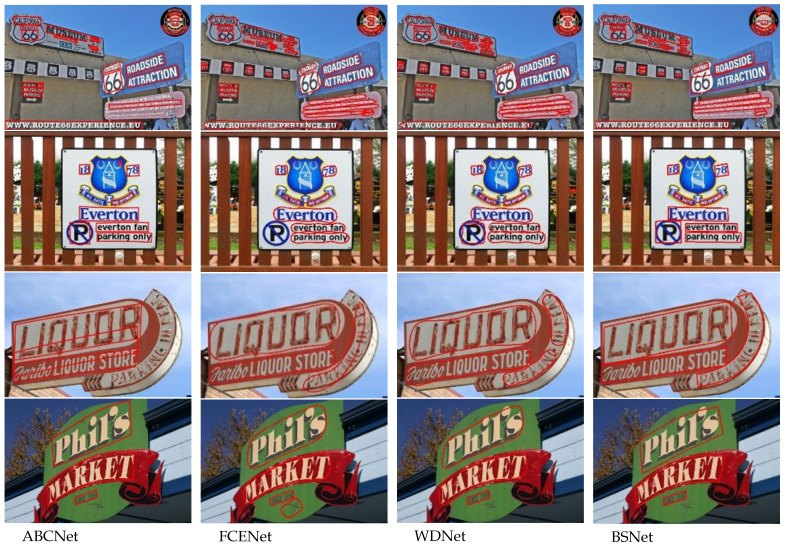
Performance comparison of the proposed model with ABCNet, FCENet, and WDNet.

**Table 1 sensors-23-02418-t001:** Ablation studies: The effect of the degree of B-Spline and the number of control points on the detection results.

B-Spline Degree	Control Points	R(%)	P(%)	F(%)
3	4	83.3	85.2	84.3
3	5	83.5	85.6	84.6
3	6	83.0	85.8	84.4
4	5	81.7	**86.8**	84.2
4	6	**83.9**	85.5	**84.7**

**Table 2 sensors-23-02418-t002:** Ablation studies: Impact of loss function on detection results on CTW1500.

$L_{reg}$	$L_{GIoU}$	R(%)	P(%)	F(%)
**✗**	**✓**	83.5	**86.0**	84.8
**✓**	**✗**	**83.9**	85.5	84.7
**✓**	**✓**	83.5	87.3	**85.4**

**Table 3 sensors-23-02418-t003:** Ablation studies: The detection results with different backbone of BSNet on CTW1500 and Total-Text.

BackBone	Params (M)	FLOPs (G)	CTW1500			Total-Text		
			R(%)	P(%)	F(%)	R(%)	P(%)	F(%)
ResNet50	39.8	122.6	83.5	87.3	85.4	82.9	87.2	85.0
ConvNeXt	40.9	125.0	86.2	87.3	**86.8**	86.4	88.8	**87.6**

**Table 4 sensors-23-02418-t004:** Comparison with previous methods on CTW1500, Total-Text, and ICDAR2015, Ext: extra training data.

Methods	Year	Ext	CTW1500			Total-Text		
			R(%)	P(%)	F(%)	R(%)	P(%)	F(%)
TextSnake [16]	2018	**✓**	85.3	67.9	75.6	74.5	82.7	78.4
PSENet [31]	2019	**✓**	79.7	84.8	82.2	84.0	78.0	80.9
CRAFT [32]	2019	**✓**	81.1	86.0	83.5	79.9	87.6	83.6
PAN [33]	2019	**✓**	81.2	86.4	83.7	81.0	89.3	85.0
Seglink++ [34]	2019	**✓**	79.8	82.8	81.3	80.9	82.1	81.5
DRRG [28]	2020	**✓**	83.0	85.9	84.5	84.9	86.5	85.7
DB [35]	2020	**✓**	80.2	86.9	83.4	82.5	87.1	84.7
ABCNetv1 [13]	2020	**✓**	78.5	84.4	81.4	81.3	87.9	84.5
ContourNet [36]	2020	**✗**	84.1	83.7	83.9	83.9	86.9	85.4
TextRay [37]	2020	**✗**	80.4	82.8	81.6	77.9	83.5	80.6
ABCNetv2 [12]	2021	**✓**	83.8	85.6	84.7	84.1	90.2	87.0
ReLaText [38]	2021	**✓**	83.3	86.2	84.8	83.1	84.8	84.0
FAST [39]	2021	**✗**	80.4	87.2	83.7	82.5	90.5	86.3
FCENet [14]	2021	**✓**	83.4	87.6	85.5	82.5	89.3	85.8
TPSNet [30]	2021	**✓**	85.1	**87.7**	86.4	84.6	**90.8**	87.6
DBNet++ [29]	2022	**✓**	**87.9**	82.8	85.3	**88.9**	83.2	86.0
WDNet [15]	2022	**✓**	84.0	87.6	85.8	82.9	87.9	85.3
LeafText [17]	2022	**✗**	83.9	87.1	85.5	84.0	90.8	87.3
Wang et al. [40]	2023	**✗**	80.5	86.1	83.6	83.4	89.6	86.4
CA-STD [41]	2023	**✓**	84.5	83.0	83.8	82.1	82.9	82.5
Ours	2023	**✗**	86.2	87.3	**86.8**	86.4	88.8	**87.6**

## Data Availability

CTW-1500 is publicly available at https://github.com/Yuliang-Liu/Cur-ve-Text-Detector, accessed on 1 January 2023. TotalText is publicly available at https://github.com/cs-chan/Total-Text-Dataset, accessed on 1 January 2023.

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
