# Peer review of "Arbitrary-Shaped Text Detection with B-Spline Curve Network"

_sensors, 2023, doi:10.3390/s23052418_

Round 1

Reviewer 1 Report

1. Numerous unexplained letter abbreviations in the annotation.

2. According to the text: FPN -? As you can further guess, this is

Feature Pyramid Networks,

«The Transformer decoder regresses the B-Spline control point coordinates of the text contours from the feature maps of the text regions. Lastly, the B-Spline decoder reconstructs the boundary coordinates of the text regions using the B-Spline control points and obtains the final detection result after removing the repeated predictions by the non-maximum suppression algorithm (NMS)."??

explain!! or give a link!

Next comes a long and confusing verbal explanation of the algorithm. Figure 2, in fact, does not explain anything.

A completely illiterate definition of a B-spline!! There's nothing to talk about!

Where is the degree, where is the recursion, where is the parameter ???

where in (4) MNC? This is the essence of redefined interpolation equations!!!

"Given a set of control points, {Qk}, k = 0, 1, . . . , n, we use cubic B-Spline curves to interpolate the set of points. We establish the following equations: » ??? a pass!!!

(7) how can n in the number of points and the degree of the spline coincide?? This is true for Bezier curves, from where, probably, everything is written off without understanding the essence of interpolation processes!!!

You mock readers - the matrix is tridiagonal, you call it triangular, there is no output of boundary conditions, etc.

D0, Dn are essentially components of the tangent vector!!! And where to get them?  From MNC? explain!! or give a link!

There are splines without boundary conditions, and they should be used in this situation!!

As a result, (8-10) looks like an interpolation problem, but the most important thing is that the NMS method remains without explanation.

What is the Hungarian algorithm? In the source: "Because we generate predictions jointly, common post-processing steps such as nonmaximum suppression are unnecessary." Some contradictions!!!

Why can't the same thing be done using a regular MNC? Why involve a neural network?

The authors should prepare a separate professional section on comparing the methods of MNC and neuropodbor to find a solution to the problem of determining the coefficients of the B-spline.

Figure 4. Example of segmentation rectangle GIoU. Here it becomes a little clearer - this is what you need to dance around, giving for example the results of other algorithms!!

But how the Segmentation rectangle is formed in this case is not explained quite intelligibly? For example, how are the horizontals and verticals drawn? A lot depends on this too!! By the way control points are not interpolation points, and what are you using really? What is the task of interpolation for?

How did the authors plan to work with 4th degree B-splines? There, when solving the interpolation problem, completely different mathematical problems arise!!

After all, it was possible to write through the usual form of the B-spline, without interweaving the interpolation equations. Perhaps this is what the authors plan to do in the next step :

"light weight will be investigated to increase its practicality"?

And it is even better to write a spline in a wavelet form, which would undoubtedly have a beneficial effect on the stability of calculations!!!

And after all there is not a single link to a serious textbook on splines and wavelets!

Reviewer 2 Report

Dear Authors,

Thank you for submitting your work to the Sensors-MDPI journal. Some improvements need to be considered to make your work better.

0.    Introduction

·        The introduction should not number 0 as a section, please revise your paper section.

·        At the end of the introduction, I suggest adding a paragraph that represents a map of the paper and briefly describes each section in one or two sentences.

1. Related works

·        This section is a bit short, so please add recent works (i.e. from 2022).

2.1. Datasets resample

- Here, please add a link/reference to the CTW1500 dataset.

3.      Experiments

Please add a paragraph that briefly describes what this section will discuss.

References: 

Please revise your references and add missing information, such as conference location like this reference “6. Redmon, J.; Divvala, S.; Girshick, R.; Farhadi, A. You only look once: Unified, real-time object detection. In Proceedings of the  Proceedings of the IEEE Computer Society Conference on Computer Vision and Pattern Recognition, 2016, Vol. 2016-Decem.  https://doi.org/10.1109/CVPR.2016.91.

Round 2

Reviewer 1 Report

No Comments and Suggestions